

# Salivary microbiomes of indigenous Tsimane mothers and infants are distinct despite frequent premastication

Cliff S. Han[1,*], Melanie Ann Martin[2,*], Armand E.K. Dichosa[1], Ashlynn R. Daughton[3], Seth Frietze[4], Hillard Kaplan[5], Michael D. Gurven[6] and Joe Alcock[7]

[1] Bioscience Division, Los Alamos National Laboratory Los Alamos, NM, USA
[2] Department of Anthropology, Yale University, New Haven, CA, USA
[3] Analytics, Intelligence and Technology (A) Division, Los Alamos National Laboratory, Los Alamos, NM, USA
[4] Department of Medical Laboratory and Radiation Sciences, University of Vermont, Burlington, VT, USA
[5] Department of Anthropology, University of New Mexico, Albuquerque, NM, USA
[6] Department of Anthropology, University of California, Santa Barbara Santa Barbara, CA, USA
[7] Department of Emergency Medicine, University of New Mexico, Albuquerque, NM, USA
[*] These authors contributed equally to this work.

Corresponding author
Joe Alcock, joalcock@salud.unm.edu

## ABSTRACT

**Background**. Premastication, the transfer of pre-chewed food, is a common infant and young child feeding practice among the Tsimane, forager-horticulturalists living in the Bolivian Amazon. Research conducted primarily with Western populations has shown that infants harbor distinct oral microbiota from their mothers. Premastication, which is less common in these populations, may influence the colonization and maturation of infant oral microbiota, including via transmission of oral pathogens. We collected premasticated food and saliva samples from Tsimane mothers and infants (9–24 months of age) to test for evidence of bacterial transmission in premasticated foods and overlap in maternal and infant salivary microbiota. We extracted bacterial DNA from two premasticated food samples and 12 matched salivary samples from maternal-infant pairs. DNA sequencing was performed with MiSeq (Illumina). We evaluated maternal and infant microbial composition in terms of relative abundance of specific taxa, alpha and beta diversity, and dissimilarity distances.

**Results**. The bacteria in saliva and premasticated food were mapped to 19 phyla and 400 genera and were dominated by Firmicutes, Proteobacteria, Actinobacteria, and Bacteroidetes. The oral microbial communities of Tsimane mothers and infants who frequently share premasticated food were well-separated in a non-metric multidimensional scaling ordination (NMDS) plot. Infant microbiotas clustered together, with weighted Unifrac distances significantly differing between mothers and infants. Infant saliva contained more Firmicutes ($p < 0.01$) and fewer Proteobacteria ($p < 0.05$) than did maternal saliva. Many genera previously associated with dental and periodontal infections, e.g. *Neisseria*, *Gemella*, *Rothia*, *Actinomyces*, *Fusobacterium*, and *Leptotrichia*, were more abundant in mothers than in infants.

**Conclusions**. Salivary microbiota of Tsimane infants and young children up to two years of age do not appear closely related to those of their mothers, despite frequent premastication and preliminary evidence that maternal bacteria is transmitted to

premasticated foods. Infant physiology and diet may constrain colonization by maternal bacteria, including several oral pathogens.

# INTRODUCTION

Premastication is the practice of feeding an infant with food chewed by its mother or other caregivers. Beginning as early as one month of age and continuing for two years or more, the practice has been reported in nearly a third of 119 traditional societies surveyed in ethnographic literature (*Pelto, Zhang & Habicht, 2010*), and is still practiced across diverse societies today (e.g., China (*Pelto, Zhang & Habicht, 2010*), South Africa (*Maritz, Kidd & Cotton, 2011*)). Premasticating mechanically processes, lubricates, and adds salivary amylase to foods, facilitating digestion and absorption (*Humphrey & Williamson, 2001*) As such, the practice may have been favored throughout much of human evolution to enhance nutritional availability of weaning foods (*Pelto, Zhang & Habicht, 2010*).

Premastication also has immunological implications. Adult saliva contains lysozyme and antimicrobial peptides that inhibit the growth of foodborne pathogens (*Ellison 3rd & Giehl, 1991*) Transmission of adult oral microbiota may promote immunotolerance and protect against development of allergic and other non-communicable diseases. (*Hesselmar et al., 2013*). Conversely, others have raised concerns that premastication may transfer HIV (*Maritz, Kidd & Cotton, 2011*; *Gaur et al., 2009*) and bacteria with cariogenic or pathogenic potential, including group A streptococcus (*Steinkuller, Chan & Rinehouse, 1992*) and *Streptococcus mutans* (*Berkowitz, Turner & Green, 1981*).

Vertical transmission of maternal bacteria is a major source of microbial colonization during early life, as demonstrated by the effect of birth mode and breastfeeding on developing infant gut microbiomes (*Funkhouser & Bordenstein, 2013*). The extent to which infant oral microbial communities are varyingly influenced by maternal and other environmental exposures, however, is less clear. For example, maternal oral community structure at birth does not appear to influence initial colonization of infant oral microbiota (*Dominguez-Bello et al., 2010*) and mothers and infants may harbor distinct oral microbial communities for at least the first year of life (*Cephas et al., 2011*; *Song et al., 2013*). These differences suggest strong developmental constraints on infant oral microbiota, as only with increasing age and dental eruption do their microbial communities become more complex and show greater overlap with those of their mothers (*Crielaard et al., 2011*; *Tanner et al., 2002a*). At the same time, maternal transmission of *S. mutans* has been detected in pre-dentulous infants, with colonization mediated by frequency of maternal-infant contact and maternal levels of the bacteria (*Tanner et al., 2002a*; *Caufield, Cutter & Dasanayake, 1993*; *Wan et al., 2001*).

Differences in maternal-infant behaviors may therefore variably influence rates of maternal bacterial transmission across and within populations. Caregiver behaviors that

can transmit salivary bacteria to infants include kissing, sharing food and utensils, and even "licking clean" infants' pacifiers (*Hesselmar et al., 2013*). We propose that premastication may transmit a relatively greater abundance and/or diversity of maternal bacteria relative to these other behaviors, however, due to the amount of saliva transferred to chewed food and the number of oral features potentially sampled during chewing.

The Tsimane are an indigenous population of forager-horticulturalists living in the Bolivian Amazon. Tsimane patterns of pathogen exposure, hygiene, diet, and reproduction contrast sharply with those observed in more Westernized populations, and are accordingly expected to influence patterns of microbial acquisition and composition. Most Tsimane lack access to clean water, electricity, plumbing, or quality medical care, and gastrointestinal and respiratory diseases are endemic (*Gurven, Kaplan & Supa, 2007*; *Martin et al., 2013*). Oral health and hygiene practices are generally poor, with the result that children frequently exhibit caries in their deciduous teeth, and adults exhibit numerous missing teeth.

The Tsimane diet consists primarily of cultivated carbohydrates supplemented with foraged game and fish, with relatively few processed foods (*Martin et al., 2012*). Almost universally, Tsimane infants are delivered vaginally at home and breastfed on demand throughout the day and night. On average they are exclusively breastfed for about 4 months and weaned at about 19 months (*Veile et al., 2014*). Mothers are the primary caregivers responsible for feeding children up through the first two years of life. Tsimane mothers do not provision infants with commercially prepared foods or cereals. In general, infants sit with their mothers during meals, and are spoon- or hand-fed small portions of whatever their mothers are eating. Before six months of age the infant diet may be primarily restricted to liquids and broths, with liquids sometime given mouth-to-mouth. Premastication is observed up through the first two years of life, and is primarily restricted to foods deemed too hot, too dry, or to pose choking hazards (e.g., stews, roasted plantains, meat, and fish).

In a recent survey on the feeding practices of 132 Tsimane infants and young children aged 0–35 months (from here on "infants"), 84% of mothers reported premasticating foods for their infants at least once, and 54% reported premasticating foods the day prior (*Martin, 2015*). Fathers and siblings were observed sharing snacks with infants on occasion, but were not observed or systematically documented to premasticate.

As preliminary evidence that premastication may influence oral bacteria transmission and microbial relatedness among mothers and infants, we studied microbiota extracted from premasticated food and matched maternal and infant saliva samples of Tsimane subjects. We hypothesized that owing to the high frequency of premastication, salivary microbiomes of related Tsimane mother–infant pairs would be more similar to one another than those of unrelated pairs. Findings from this study may advance understanding of infant oral microbial development in non-Western contexts. Results may also have direct implications for Tsimane infant health, as on the one hand premastication may buffer against nutritional losses and pathogen exposure associated with weaning transitions, but on the other may increase already high risks of oral pathogen transmission and associated dental diseases.

**Table 1** Characteristics of sampled mothers and infants.

| Subject | Sample | Sex | Village | Age (years) | # Teeth | # Caries | Sampling method | Most recent premasticated foods documented |
|---|---|---|---|---|---|---|---|---|
| Infant.1 | BSA1 | F | 1 | 1.3 | 8 | 0 | Buccal swab | 1 day prior: fish, meat stew, plantain |
| Infant.2 | BSA2 | F | 1 | 1.1 | 6 | 0 | Buccal swab | 1 day prior: rice, plantain, meat stew |
| Infant.3 | BSA3 | F | 1 | 0.8 | 4 | 0 | Buccal swab | 1 day prior: plantain, meat |
| Infant.4 | BSA4 | M | 1 | 1.5 | 14 | 0 | Buccal swab | 1 day prior: fish stew |
| Infant.8 | BSA8 | M | 2 | 2.0 | 16 | 1 | Buccal swab | 1 day prior: fish |
| Infant.9 | BSA9 | F | 2 | 1.7 | 15 | 8 | Buccal swab | 1 day prior: fish |
| Infant.10 | BSA10 | M | 2 | 1.7 | 16 | 0 | Buccal swab | N/A |
| Infant.11 | BSA11 | M | 2 | 1.1 | 8 | 0 | Buccal swab | N/A |
| Infant.12 | BSA12 | M | 3 | 0.8 | 1 | 0 | Buccal swab | 11 days prior: rice, plantain |
| Infant.13 | BSA13 | M | 3 | 1.2 | 10 | 0 | Buccal swab | 4 days prior: rice |
| Infant.14 | BSA14 | M | 3 | 1.1 | 9 | 0 | Buccal swab | 10 days prior: plantain, meat, fish stew |
| Infant.15 | BSA15 | M | 3 | 0.9 | 5 | 0 | Buccal swab | 10 days prior: stew, plantain, meat |
| Mother.1 | MSA1 | F | 1 | 20.1 | 22 | 1 | Buccal swab | |
| Mother.2 | MSA2 | F | 1 | 18.8 | 24 | 0 | Buccal swab | |
| Mother.3 | MSA3 | F | 1 | 36.3 | 18 | 0 | Buccal swab | |
| Mother.4 | MSA4 | F | 1 | 21.3 | 27 | 1 | Buccal swab | |
| Mother.8 | MSA8 | F | 2 | 29.6 | 22 | 1 | Buccal swab | |
| Mother.9 | MSA9 | F | 2 | 22.7 | 30 | 0 | Buccal swab | |
| Mother.10 | MSA10 | F | 2 | 24.6 | 24 | 1 | Buccal swab | |
| Mother.11 | MSA11 | F | 2 | 26.8 | 19 | 1 | Buccal swab | |
| Mother.12 | MSA12, FD12 | F | 3 | 24.9 | 32 | 1 | Expectoration | |
| Mother.13 | MSA13 | F | 3 | 27.7 | 15 | 8 | Expectoration | |
| Mother.14 | MSA14, FD14 | F | 3 | 31.6 | 25 | 1 | Expectoration | |
| Mother.15 | MSA15 | F | 3 | 25.2 | 21 | 1 | Expectoration | |

**Notes.**

Subjects from the same family have the same indexing number.

BSA, infant saliva; MSA, mother saliva; FD, premasticated food.

'Most recent premasticated foods' were documented from subjects' most recent 24-hour dietary recalls. Dietary recall was not collected for subjects BSA10 and BSA11, though at sample collection their mothers reported still premasticating all types of foods. Premasticated food sample FD14 was collected approximately 40 min after cooking, and was chewed for 17 s. Sample FD12 was collected 50 min after cooking, and was chewed for 8 s.

## METHODS

### Study design

Maternal and infant saliva samples were collected from 12 dyads in three villages. The participating dyads were part of an ongoing research project being conducted by Author 2 on Tsimane infant feeding transitions and health outcomes. All infants were reported by their mothers to still receive premasticated foods. For 10 of 12 infants, premastication in the two weeks prior to sample collection was also documented in routinely collected 24-hour dietary recalls (Table 1). All infants were currently breastfeeding, though measures of daily breast milk intake were not collected or estimated. Neither maternal nor infant subjects had taken antibiotics in the week prior to sample collection. Subjects were selected as a matter of convenience, as in their respective communities they lived in close proximity to the researcher's field base.

Mothers and infants varied in their respective ages and dental health (Table 1). The following maternal subjects were sisters: Mother.1/Mother.3; Mother.10/Mother.11; Mother.13/Mother.15.

The four mothers from Village 3 who participated in the controlled experiment to collect premasticated food samples were also neighbors and frequently exchanged food or shared meals. As no dental records were available, we report the total number of teeth per subject rather than missing teeth. We assume that incomplete tooth counts for adult subjects (less than 32 teeth) primarily represent missing teeth due to dental disease and/or manual extraction, whereas incomplete tooth counts in infants (less than 20) primarily reflect different stages of tooth eruption.

## Sample collection
### Saliva

Twenty-four saliva samples were collected from 12 Tsimane mother–infant pairs from three different villages (Table 1). Saliva samples from subjects in Villages 1 and 2 were collected by placing a sterile buccal cell collection swab (Catch-All$^{TM}$ Sample Collection Swab; Epicentre$^®$) on the subject's tongue for five seconds. The swabs were kept in the sterile collection tubes at ambient temperature and transported back to a field laboratory. The buccal swabs were transferred to cryotubes and stored in liquid nitrogen within approximately 30 min of sample collection.

Saliva and premasticated food samples were collected from participants in Village 3 only, with saliva samples collected prior to food consumption. Infant saliva samples in this group were collected using the buccal cell collection method described above. The maternal saliva samples were collected via expectoration rather than buccal swab in order to duplicate the sampling method used to collect premasticated food. Mothers were asked to expectorate into disposable plastic cups, and the researcher transferred the saliva on site into individually pre-labeled 2 ml cryotubes using disposable plastic pipettes.

### Premasticated food

A traditional Tsimane stew ("*jo'na*," made from green plantains and meat) was prepared by one of the participants in her home, with ingredients and cookware supplied by Author 2 and the participant. All cookware and utensils were cleaned prior to use with water and a 70% alcohol solution. The stew was made by boiling approximately 1 kg of dried beef, 20 unripe plantains, and vegetable oil and salt to taste. Cooking water was sourced from a local river, and the stew was boiled for approximately 19 min over an open flame.

Five control samples of unchewed food were taken from the pot approximately 17 min after the pot was removed from heat. The attending author served food to and gathered samples from each of the four participants. Mothers were instructed to premasticate food just as they normally would for their infants, and then expectorate the premasticated bite into a disposable plastic cup. The unchewed and premasticated food samples were transferred into pre-labeled 2 ml cryotubes using wooden oral tongue depressors. The samples were kept in an insulated bag at ambient temperature during the sampling collection process, and then returned to the field site laboratory for immediate storage in liquid nitrogen. The time between sample collection and storage for all saliva and food

samples from Village 3 ranged from approximately 1–1.75 h. All samples were transported on dry ice to the US and stored at −80 °C until analysis.

## DNA isolation

Following manufacturers' protocols, a PSP Saliva Gene Kit (STRATEC Molecular GmbH, Berlin, Germany) was used to extract DNA from the saliva samples with added lysozyme for better lysis of Gram positive bacterial cells as suggested by the supplier. And a PowerLyzer® UltraClean® Microbial DNA Isolation Kit (MoBio Cat# 12255-50; MO BIO Laboratories, Carlsbad, CA, USA) was used to extract DNA from the food samples. DNA concentrations were measured with a Qubit Fluorometer (Life Technologies, Grand Island, NY, USA). Only two of the five control food samples were selected for DNA analysis, and did not yield sufficient DNA for PCR amplification. Two of the four premasticated food samples (corresponding to Mother.13 and Mother.14, Table 1) also failed to yield sufficient DNA for further analysis. PCR amplification and sequencing was therefore limited to the 24 saliva samples and only two premasticated food samples (FD12 and FD14, Table 1).

## PCR amplification and sequencing

The V4 region of the bacterial 16S rRNA gene was amplified with a NEXTflex 16S rRNA gene V4 Amplicon-Seq Kit (with 48 barcodes) (BIOO Scientific, Austin, TX, USA) following the manufacturer's instructions. The kit produces amplicons of 250 bases that can be sequenced with Illumina technology from both end with significant overlap. The PCR primers are V4_515f adaptor 5′AATGATACGGCGACCACCGAGATCTACACTAT GGTAATTGT GTGCCAGCMGCCGCGGTAA3′ and V4_806r adaptor 5′CAAGCAGAA GACGGCATACGAGAT XXXXXXXXXXXX AGTCAGTCAGCC GGACTACHVG GGTWTCTAA3′, with "X" indicating indexing sequencing capable to code up to 1,152 individual samples. Sequences at 5′ ends of the primers are adaptors for Illumina sequencing, those at 3′ ends are 16S rRNA gene primers. In total, 50 ng DNA was used as a template for each reaction. The amplified DNAs were verified on a gel and pooled at an equal molar ratio. The pooled DNA samples were purified with Agencourt AMPure beads. Sequencing was performed on a MiSeq (Illumina, San Diego, CA, USA).

## 16S rRNA gene sequence data processing

An analysis of the 16S rRNA gene sequencing data was performed using the mothur software package (*Schloss et al., 2009*). Mothur was selected for ease of installation, clear and brief instructions, and compatibility with other software (*Nilakanta et al., 2014*). Initially, the paired end sequences were assembled together, and any sequences with ambiguous bases (32% of them) were removed. To ease the sequence classification process, unique sequences were collected. The collection was further reduced by choosing representatives of near-identical sequences with fewer than 2 different bases. Chimeric sequences (∼18%) were removed with UCHIME (*Schloss, Gevers & Westcott, 2011*). Sequences left were then aligned using the SILVA databases supplied in mothur, and those sequences classified as Chloroplast, Mitochondria, Archaea, Eukaryota, or Unknown were removed (293 of them). The sequence yield and indexing codes for all samples analyzed are given in Table S1.

The unique sequence collection was clustered into operational taxonomic units (OTUs), with a similarity of 97%. OTU with single or double sequences were removed. Each OTU was assigned a taxonomic classification at all levels, from phylum to genus, using the reference Ribosomal Database Project (RDP) database provided in mothur. Alpha diversity was analyzed with Rarefraction curve deduced with tools from mothur. The count table was normalized with cumulative count scaling implemented in metagenomeSeq (*Paulson et al., 2013*). Distance ordination was done with normalized data. Total-sum scaling was followed for taxa comparison. The dissimilarity distances between samples were calculated in mothur using the dist.shared command. Thetayc distance matrices were used to compare phylogenetic distances among infants and mothers, and to examine differences in phylogenetic distance within and between dyads (*Yue, Clayton & Lin, 2001*) Weighted Unifrac distance metrics (*Lozupone et al., 2011*) were calculated to compare the microbial communities of mothers and infants, which result is also confirmed with ANOSIM (*Anderson, 2001*). Distance ordination was ploted with non-metric multidimensional scaling method that implemented in metagenomeSeq (*Paulson et al., 2013*). Finally, we attempted to identify some OTUs of interest at the species level using the HOMD library with the online 16S rRNA gene sequence identification service (*Chen et al., 2010*).

## Statistical analyses

Student $t$-tests were used to compare alpha diversity, abundances at different phylogenetic levels, and phylogenetic distances between mothers and infants. **The Shapiro–Wilk test was used to test the normality of the distribution of taxonomic data (the abundance of most common taxa). Of 91 data sets (12 samples each, infant and mother considered separately) 57% were normally distributed ($p < 0.05$). Mann–Whitney $U$ tests were used for data sets that were not normally distributed.** $P$-values were further adjusted for multiple comparisons using the false discovery rate method (*Benjamini & Hochberg, 1995*). All analyses were performed in R. Food samples were not statistically evaluated since only two samples were sequenced.

## Ethics approval and consent

This study was approved by the Human Subjects Committee of the University of California Santa Barbara (ANTH-GU-MI-010-3U) and the Human Subject Research Review Board of Los Alamos National Laboratory (LANL-14-11 X). The Tsimane Health and Life History Project maintains annual agreements with the local municipal government of San Borja, the Hospital of San Borja, and the Tsimane governing council, El Gran Consejo Tsimane, to conduct research with the Tsimane. Maternal subjects gave verbal informed consent for themselves and their infants to participate prior to sample collection.

## RESULTS

### General characteristics of Tsimane mothers' and infants' oral microbiomes

The bacteria in the saliva and premasticated food samples belonged to 18 different phyla and 341 genera (Table S2). The main bacterial phyla were Firmicutes (with the predominant

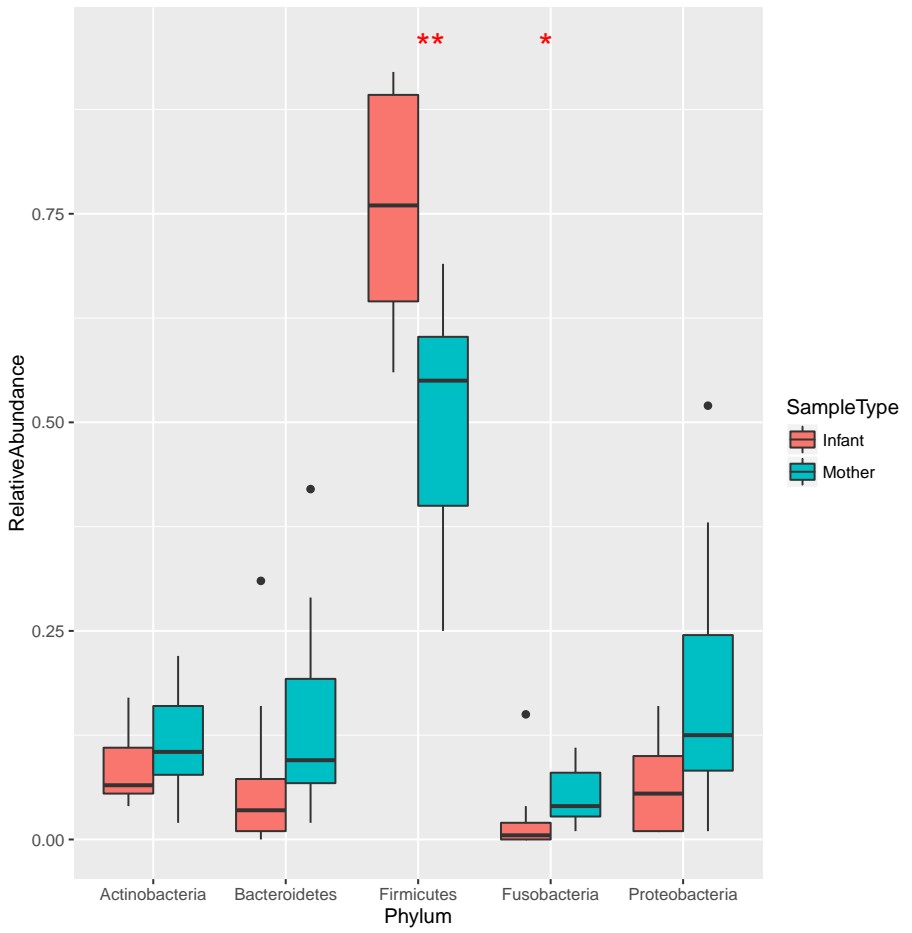

**Figure 1** **Relative abundance of the major bacterial phyla in mother and infant saliva samples.** Differences between groups of mother and infants for phyla Firmicutes and Fusobacteria are significant (**$p < 0.01$, *$p < 0.05$).

genus *Streptococcus*), Proteobacteria (*Neisseria*), Actinobacteria (*Actinomyces*), and Bacteroidetes (*Prevotella)*. Other phyla included Fusobacteria, Spirochaetes, TM7, Tenericutes, SR1, and Verrucomicrobia. The relative abundance of each phylum was different between the saliva and food samples of mothers and infants (Figs. 1 and 2). Infant saliva samples contained more Firmicutes ($p < 0.1$) and less Fusobacteria and Spirochaetes ($p < 0.5$) than did maternal samples. The mean relative abundance of the most abundant families across all saliva samples are listed in Table S3.

Thirteen genera—*Streptococcus, Neisseria,* unclassified *Pasteurellaceae, Gemella, Prevotella Leptotrichia , Actinomyces, Rothia, Veillonella, Sphingomonas,* and unclassified genera from the *Leptotrichiaceae, Flavobacteriaceae,* and Lactobacillales families—were observed in all saliva samples. The most common genus, *Streptococcus*, has 4 OTUs (OTU0001, OTU0003, OTU0029 and OTU0084) and was dominant in 92% (22/24) of saliva samples. OTU0001 is identical to *Streptococcus mitis, S. oralis, S. infantis* and *S. mitis* bv.2based on comparison with the Human Oral Microbiome Database (HOMD) (*Chen et al., 2010*) We could not distinguish among these species, as the V4 regions of their 16S

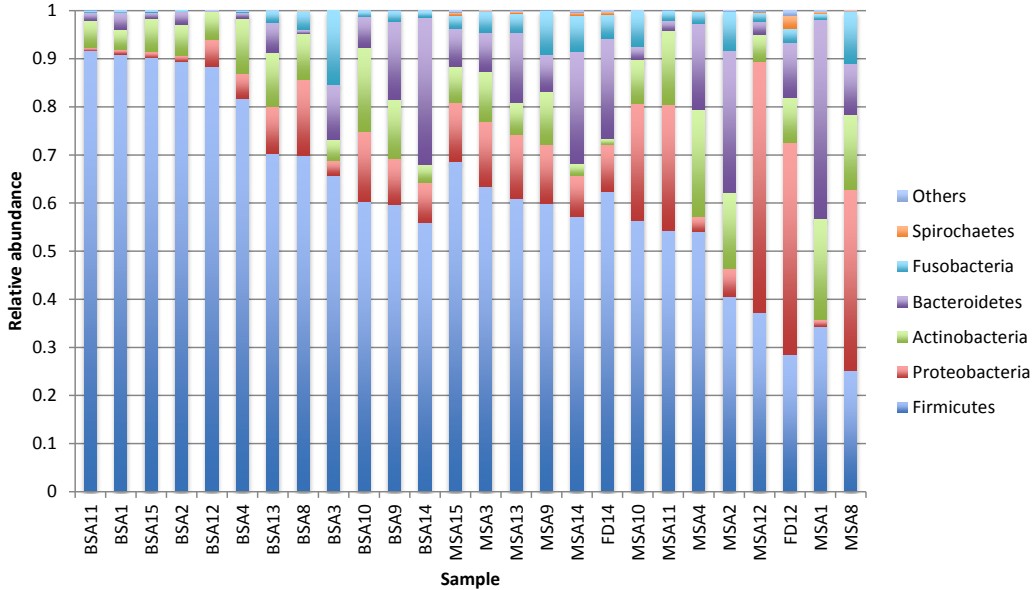

**Figure 2** **Relative abundance of the major bacterial phyla of all saliva and premasticated food samples, sorted between infant and maternal subjects.** BSA, infant saliva; MSA, mother saliva; FD, premasticated food; family index numbers are the same for infant, mother and food samples.

rRNA gene sequences are identical. It is possible that samples harbored more than one of these species. The average abundance of OTU0001 was 413% in infants' saliva and 25.2% in mothers' saliva (adjusted $p < 0.01$). OTU0003 is identical to 16S rRNA gene sequences of *S. salivarius* and *S. vestibularis* in HOMD. This OTU was about a third as abundant as OTU0001, and twice as abundant in infants (221%) as in mothers (11.5%) ($p < 0.01$). OTU0029 is identical to *S. Intermedius*. This OTU was more abundant in mothers' saliva (0.8% vs 0.1% in infants), but the difference is not statistically significant. OTU0084 matches Streptococcus sp. oral taxon 487 in HOMD at 98.8% and was found in only two infant and four maternal samples. Overall, infants' saliva contained significantly more *Streptococcus* (65.0% vs 36.9%, $p < 0.01$) and less *Leptotrichia* (0.2% vs 2.4%, $p < 0.01$) and Catonella (2 infants and 11 mothers have it, $p < 0.01$) than mothers' saliva after adjusting for false discovery rate.

## Microbial populations of premasticated foods in relation to donor salivary populations

Bacterial populations from the two premasticated food samples that could be sequenced were more similar to those of their donors than to those of other mothers, although this finding only trends toward statistical significance ($p = 0.065$), likely due to the small sample size. Differences in dissimilarity distances of the foods samples may also reflect individual differences in chewing "style" and duration. The sample that was chewed for 17 s had a smaller dissimilarity distance than that of the sample chewed for 8 s (0.061 vs 0.191), suggesting more donor bacteria may be incorporated in food with increased chewing time.

Despite within-subject similarities between premasticated food and donor saliva, 16S rRNA gene sequencing revealed distinct differences between chewed food and saliva samples

(Table S4), which were both collected via expectoration. Sixty OTUs were enriched more than 5-fold in premasticated food compared with saliva samples, and 36 OTUs were only observed at >0.01% in premasticated food samples. Some bacteria in the food samples may have been incorporated through the surface water used in food preparation, as we identified bacteria associated with plants, soil, and water reservoirs in the premasticated food samples. However, we did not yield sufficient DNA from the original unchewed food for 16S rRNA gene sequencing in two attempts, suggesting cooking decreased the availability of bacteria or bacterial DNA in the food samples.

### Relatedness of maternal and infant salivary microbiomes

We assessed relatedness of maternal and infant salivary microbiota by comparing alpha (intrasample) diversity and beta (intersample) diversity among saliva samples.

The internal sample alpha diversity was measured by rarefaction analysis (Fig. S1). On average, infant saliva samples had 41 OTUs when sampling 3148 sequences, whereas maternal samples had 87 ($p < 0.01$). The lower diversity of infants' saliva paralleled greater dominance of *Streptococcus*. Intra-sample diversity was not correlated with age or number of teeth when analyzing mothers and infants separately, but was correlated with age when analyzing the two groups together ($r = 0.75$, $p < 0.01$). Alpha diversity differed among maternal subjects according to sample collection method: $113 \pm 24$ for expectorated samples and $75 \pm 22$ for samples collected via buccal swab ($p = 0.01$).

Both ANOSIM and weighted Unifrac analysis indicated that mothers and infants had different salivary microbial community structures ($p < 0.01$). However, the difference was not significant when calculated with unweighted Unifrac analysis, which indicated that the abundance of shared OTUs contributed more to the weighted Unifrac calculation. The average distances among infant saliva samples ($0.30 \pm 0.18$) were significantly shorter than those among maternal samples ($0.48 \pm 0.21$) ($p < 0.01$), indicating greater similarity among infant microbial communities than among maternal communities (Fig. 3). Average distances calculated for related mother–infant pairs (0.45) were not significantly shorter than those of unrelated mother–infant pairs (0.47), suggesting community structures of related pairs were no more similar than those of unrelated pairs (Fig. 3). A non-metric multi-dimensional scaling ordination (NMDS) plot made using a Thetayc tree further shows that infants' salivary microbiomes were more similar in community structure to one another than to maternal microbiomes (Fig. 4).

We also calculated the average distance between each infant with all possible combinations of mothers. Across infants, this average distance was negatively correlated with infant age ($r = -0.64$, $p < 0.05$, Fig. S3A). The average distance between each maternal sample and all infant samples, however, was not associated with maternal age ($r = -0.33$, $p = 0.29$) (Fig. S3B) Tsimane salivary microbial communities may therefore become more adult-like during late infancy (9–24 months), but there is little age-associated variation in community structure during adulthood (18–41 years) in this study group.

### Oral pathogens identified in maternal and infant saliva samples

Chronic dental infections are common in Tsimane people, as evidenced by numerous caries and tooth loss in our maternal subjects (Table 1). We searched for pathogens associated with

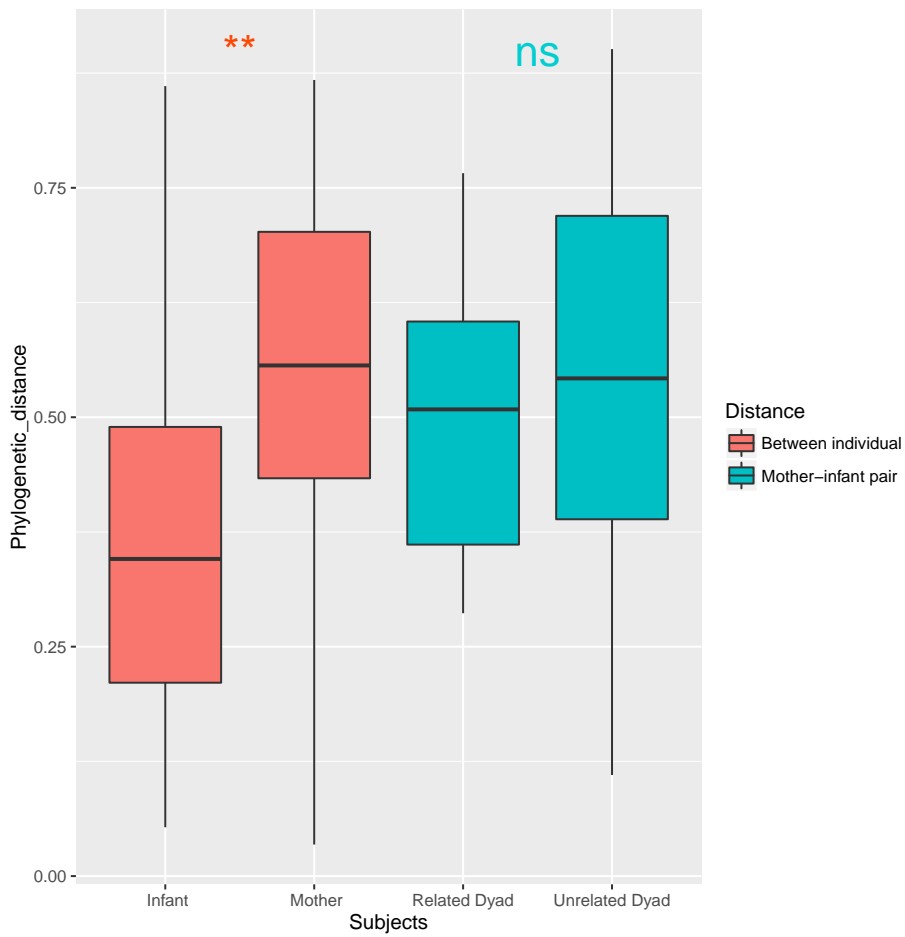

**Figure 3  Average similarity distance of group of infants, mothers, related mother-infant dyads and unrelated woman and infant pairs.** **, $p < 0.01$; ns, not significant, $p > 0.05$.

dental disease in maternal and infant samples, which could indicate vertical transmission. Using the HOMD library, OTU0120 matched *S. mutans* at 100%; OTU0143, *T. denticola* at 98%; OTU0068, *T. forsythia* at 97.2%; and OTU0033, *P. gingivalis* at 99.2% Because OTU0068 and OTU0143 matched target species at a relatively low level, we restricted our analyses to OTU0033 (*P. gingivalis*) and OTU0120 (*S. mutans*) and traced them across mother–infant dyads. However, these species-level identifications are limited because we did not perform confirmatory tests to identify these OTUs as pathogens.

In normalized results, OTU0033 (*P. gingivalis*) was observed in 11 mothers' saliva samples (92%, MSA1, MSA8, MSA12, MSA13, MSA15), but not in any of the infants' samples ($p < 0.01$). OTU0120 (*S. mutans*) was observed in 7 infants' saliva samples (BSA1, BSA11, BSA12, BSA13, BSA14, BSA15, BSA8) and 8 mothers' samples (MSA1, MSA10, MSA11, MSA12, MSA13, MSA14, MSA15, MSA9). The frequency of *S. mutans* did not significantly differ across maternal and infant samples ($p = 0.24$). Three pairs of mother–infant samples (BSA8 and MSA8, BSA9 and MSA9, BSA10 and MSA10) did not match each other in *S. mutans* status, and the remaining pairs (9) did. The occurrence

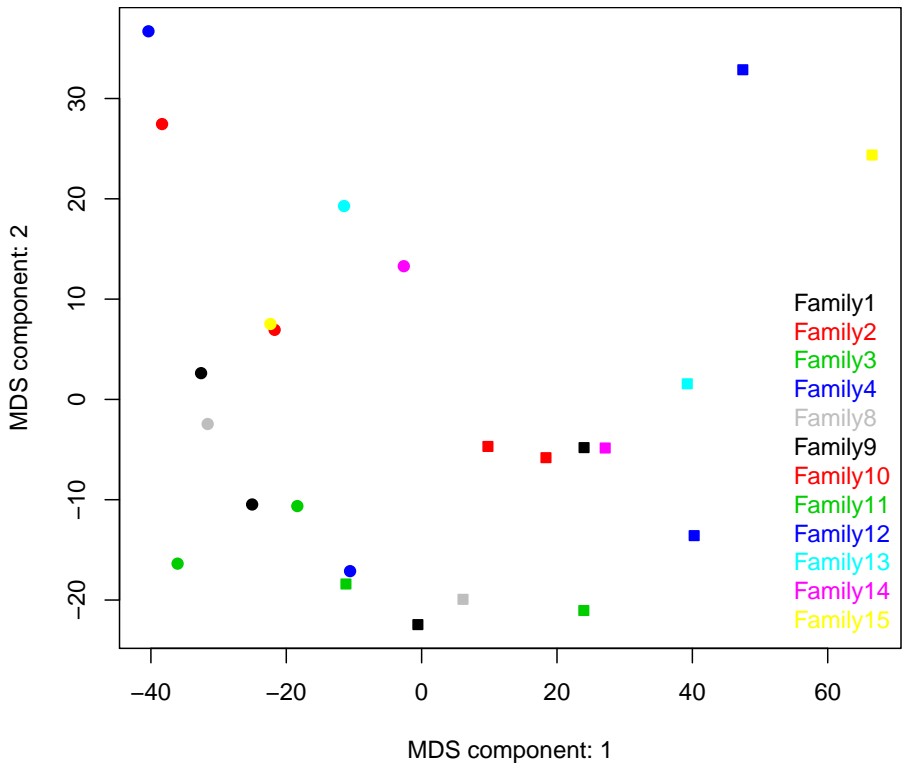

**Figure 4** **Two dimensional NMDS plot. Infant salivary microbiomes are closer to those of other infants than to their mothers in most cases.** Round, infants; square, mothers. Stress = 0.054911.

and abundance of the two possible pathogens were not correlated with age, the number of teeth, dental caries, or the loss of teeth within each group. There were 754 *S. mutans* reads from infant saliva sample BSA14, compared with fewer than 5 in other samples, but the subject providing this sample had no observable dental caries at the time of sampling.

Fourteen of 22 genera associated with gingivitis were identified in 50% or more of maternal samples (Table S5). Seven of these 14 genera were identified in only 1/3 or less of infant samples, and no *Eubacterium* or *Johnsonella* were identified in any infant sample (Table S5). *Fusobacterium, Leptotrichia, Porphyromonas,* and *Prevotella* were identified in at least 83% of both maternal and infant samples (Table S5). All were more abundant in the maternal samples (Fig. S2). With the exception of *Tannerella,* the gingivitis-associated OTUs positively identified in infant samples were also identified in the corresponding maternal samples, but not vice versa.

## DISCUSSION

Maternal-offspring transmission of salivary bacteria through premastication may have been universal in ancestral human populations and remains common today (*Pelto, Zhang & Habicht, 2010*). This is the first study to document the microbiome of premasticated foods and to examine relationships between maternal and infant salivary microbiota in an indigenous population with frequent sharing of pre-chewed food.
Overall, the patterns of oral microbial composition observed for the Tsimane support existing findings for mothers and infants in diverse populations. The five most predominant phyla observed for Tsimane subjects (Fig. 1) were the same as those previously observed in a sample of US children aged 3–18 (*Crielaard et al., 2011*) and in a sample of US mothers and their infants aged 3–6 months old (*Cephas et al., 2011*), Across all three study populations, Firmicutes was the most abundant phylum, with *Streptococcus* predominating samples and found at higher relative abundance in infants and children than in adult subjects. The relative abundance of Bacteroidetes in samples from Tsimane children aged 9–24 months was lower as compared to that of older US children but higher compared to that of younger US infants, which may support the observation that this phylum increases with age (*Cephas et al., 2011*). Dissimilarity distances across Tsimane infant and maternal samples also decreased with age (Fig. S3), consistent with previous observations that oral microbial composition changes quickly during childhood (*Stahringer et al., 2012*), likely due to changes in dentition and diet. The most predominant 13 genera observed among Tsimane samples are also consistent with the previous observation of the existence of a core oral microbiome (*Zaura et al., 2009*; *Lazarevic et al., 2013*).

The bacteria extracted from the premasticated food samples in our study appear to have been at least partially derived or modified by chewing. Mastication has been shown to release Acinetobacter (*Slots et al., 1991*) and *Staphylococcus* (*Cuesta et al., 2010*) from the subgingival pocket. The chewed food samples in this study harbored a greater diversity of bacteria than corresponding subject saliva samples, which may reflect contact with dental, lingual, buccal, and subgingival surfaces during chewing. This finding tentatively supports our proposal that premastication may transmit a greater diversity of oral bacteria—including potential pathogens—than more superficial modes of salivary exchange, including non-romantic kissing and utensil-sharing However, our inferences are confounded by the fact that DNA from saliva and premasticated food were isolated using different kits (*Lazarevic et al., 2013*), which were selected to optimize the bacterial DNA yield respective to the sample sources. This limitation can be remedied by using consistent collection and laboratory protocols in future studies.

Despite evidence that maternal bacteria is incorporated into premasticated food, salivary microbiomes of related mother–infant pairs known to share premasticated foods were no more similar than those of unrelated pairs. These results are in line with existing cross-cultural research showing distinct clustering between mothers and infants with respect to both gut (*Palmer et al., 2007*; *Koenig et al., 2011*; *Yatsunenko et al., 2012*; *Sampaio-Maia & Monteiro-Silva, 2014*) and oral microbiomes (*Dominguez-Bello et al., 2010*; *Cephas et al., 2011*). Evidence of transmission of oral pathogens between Tsimane mothers and infants was also weak. The abundance of *Actinomyces*, *Leptotrichia*, and *Fusobacterium*—previously associated with dental, periodontal, and other infections (*Eribe & Olsen, 2008*; *Fujinaka et al., 2013*; *Jiang et al., 2014*) —was significantly lower for infants as compared to their mothers. Bacteria identified as *S. mutans* occurred in two infants of mothers who harbored the bacteria, but also in three infants of mothers who did not.

Ultimately, several factors may limit the influence of premastication on Tsimane infant salivary microbiomes, as well as our ability to detect these influences in our study. First,

we do not have precise data on the frequency of premastication prior to sampling or over time in our subjects. However, even frequent maternal salivary transfer via premastication may have only transitory effects on infant microbial communities, as is suggested by a recent study of bacterial transmission following intimate kissing in adults (*Kort et al., 2014*). Second, while infants are continually exposed to novel environmental bacteria, acquisition of particular species and overall community structure in early life appears tightly constrained by developing features of the oral cavity (*Cephas et al., 2011*; *Klein et al., 2004*; *Gizani et al., 2009*). Infants in our sample exhibited various stages of dental eruption, which may have limited opportunities for colonization by maternal bacteria. Acquisition of *S. mutans*, for example, may occur during a well-delineated "window of infectivity" corresponding with tooth eruption (*Caufield, Cutter & Dasanayake, 1993*), though *S. mutans* has been detected in predentulous infants, and at higher proportions among infants of infected vs. non-infected caregivers (*Tanner et al., 2002b*).

Finally, intensive breastfeeding in this population may be a significant factor shaping infant oral microbial composition, oral pathogen transmission and disease risk. Breast milk contains maternally-derived bacteria and nutritional and antimicrobial factors that may uniquely influence or interact with oral bacteria to shape oral community composition in breastfed infants (*Holgerson et al., 2013*; *Habicht & Pelto, 2016*). For example, certain isolates of *S. oralis, S. sanguis, and S. mitis* have been shown to produce IgA1 protease in neonates, which given abundant IgA in breast milk, may confer an ecological advantage over other bacteria that lack this enzyme (*Cole et al., 1994*) *In vitro* studies have also demonstrated that breastmilk mixed with neonatal saliva stimulates hydrogen peroxide to selectively inhibit the growth of *S. aureus* and *Salmonella* spp (*Al-Shehri et al., 2015*).

Though some research suggests breastfeeding increases risks of early childhood caries—particularly when breastfeeding is prolonged after one year of age and includes night nursing. *Salone, Vann & Dee (2013)*—evidence for this association is inconsistent and frequently confounded by undocumented variability in infant diets, oral hygiene, and maternal oral health (*White, 2008*; *Tham et al., 2015*). It is of particular interest then that in this study caries were observed in only two infant subjects, though all (at up to two years of age) were still breastfeeding and co-sleeping with their mothers. That is, we have no evidence that these behaviors increase disease risk in the context of Tsimane nutritional and disease ecologies. Similarly, caries can occur in the absence of specific associated pathogens such as *S. mutans* (*Aas et al., 2005*) and ultimately many bacterial species, or the balance of particular species, play a role in the etiology of caries (*Gross et al., 2012*; *Kanasi et al., 2010*; *Luo et al., 2012*). Frequent transmission of disease-associated bacteria through premastication, therefore, may not be sufficient to induce disease, as such bacteria may be "pathogenic" only in the context of the habitat offered by their host.

The Tsimane do have generally poor oral health, as is evident from the number of caries and missing teeth observed in maternal subjects, and which we have also observed in older children and adolescents. With increasing age and dental maturation, Tsimane infant oral microbiomes become more adult like, while the frequency of breastfeeding and premastication gradually decline (*Martin, 2015*; *Veile et al., 2014*). Younger children, therefore, may be protected by prebiotic and immunological constituents in breast milk and

maternal saliva that selectively promote the growth of some commensals while inhibiting establishment of pathogens. Additional research is needed, however, to identify population-specific risks of oral pathogen transmission and related disease etiologies, as well as to draw firm conclusions about the relative costs of disease exposure vs. potential nutritional or immunological benefits from premastication (*Habicht & Pelto, 2016*). Comparative studies, larger sample sizes, and more precise measures of premastication frequency, breastfeeding intensity, and dietary composition are also necessary to better understand how these behaviors intersect in shaping infant oral microbial development and disease risk at different stages of dentition.

## CONCLUSION

Oral microbiomes of Tsimane infants and young children up to two years of age do not appear closely related to those of their mothers, despite frequent premastication and evidence that maternal oral bacteria is transmitted to premasticated foods. Tsimane mothers displayed poor oral health and abundant oral pathogens, but these bacteria were found less frequently in infant samples and did not systematically co-occur within dyads. Our conclusions are limited by a small sample size and two methodological inconsistencies that may have influenced results: differing methods of salivary sample collection for maternal subjects, and the use of different DNA kits for salivary and food samples. Future research should avoid these limitations, simultaneously examine oral microbial composition in a comparative sample of infants who do not receive premasticated foods, and consider the confounding influence of factors such as the stage of dental eruption and intensity and duration of breastfeeding.

Despite these limitations, the patterns of salivary microbial composition observed for Tsimane mothers and infants—in terms of most abundant phyla and pattern of maternal and infant clustering—are broadly consistent with those previously observed for mothers and young infants in the US (*Cephas et al., 2011*). However, we did observe differences in the relative abundance of distinct phyla and genera across these two sample populations, which may reflect differences in pathogen exposure, diet, and oral hygiene. The current study thus highlights the need for broad sampling in human microbiome research. Cross-cultural studies of populations with diverse diets, disease ecologies, behaviors, and socioeconomic conditions are especially informative in improving our understanding of how human oral microbiomes are acquired and shaped across the lifespan.

## ACKNOWLEDGEMENTS

We thank David Sela and anonymous reviewers for their helpful critiques on earlier drafts of the manuscript. We are grateful for the support of Tsimane participants and the staff of the Tsimane Health and Life History Project in Bolivia.

### Funding

This project is supported by Los Alamos National Laboratory through Laboratory Directed Research and Development 20110034DR (Author 1); NSF Doctoral Dissertation Grant BCS-1232370 and Wenner-Gren Dissertation Fieldwork Grant (Author 2); NIH/NIA R01AG024119-01 (Authors 5 and 6). The funders had no role in study design, data collection and analysis, decision to publish, or preparation of the manuscript.

### Grant Disclosures

The following grant information was disclosed by the authors:
Los Alamos National Laboratory: 20110034DR.
NSF Doctoral Dissertation: BCS-1232370.
Wenner-Gren Dissertation Fieldwork: NIH/NIA R01AG024119-01.

### Competing Interests

The authors declare there are no competing interests.

### Author Contributions

- Cliff S. Han conceived and designed the experiments, performed the experiments, analyzed the data, contributed reagents/materials/analysis tools, wrote the paper, prepared figures and/or tables, reviewed drafts of the paper, contributed equally.
- Melanie Ann Martin conceived and designed the experiments, performed the experiments, analyzed the data, wrote the paper, prepared figures and/or tables, reviewed drafts of the paper, contributed equally.
- Armand E.K. Dichosa performed the experiments, analyzed the data, prepared figures and/or tables, reviewed drafts of the paper.
- Ashlynn R. Daughton and Seth Frietze performed the experiments.
- Hillard Kaplan conceived and designed the experiments, reviewed drafts of the paper.
- Michael D. Gurven conceived and designed the experiments, wrote the paper, reviewed drafts of the paper.
- Joe Alcock conceived and designed the experiments, analyzed the data, wrote the paper, prepared figures and/or tables, reviewed drafts of the paper.

### Human Ethics

The following information was supplied relating to ethical approvals (i.e., approving body and any reference numbers):

UCSB IRB Approval number: ANTH-GU-MI-010-3U

Los Alamos National Laboratory IRB Identification Number: LANL-14-11 X.

This study was approved by the Human Subjects Committee of the University of California Santa Barbara and the Human Subject Research Review Board of Los Alamos National Laboratory. The Tsimane Health and Life History Project maintains annual agreements with the local municipal government of San Borja, the Hospital of San Borja, and the Tsimane governing council, El Gran Consejo Tsimane, to conduct research with

the Tsimane. Maternal subjects gave informed consent for themselves and their infants to participate prior to sample collection.

## Data Availability

Sequence data was submitted to Sequence Read Archive at National Center for Biotechnology Information and available under Bioproject Accession PRJNA320081 with Group Accession Number SRP074186.

## Supplemental Information

Supplemental information for this article can be found online at http://dx.doi.org/10.7717/peerj.2660#supplemental-information.

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
