# Peer review of "Salivary microbiomes of indigenous Tsimane mothers and infants are distinct despite frequent premastication"

_PeerJ, doi:10.7717/peerj.2660_

## Round 0.1 · original submission · Minor Revisions

As you will see, all three reviewers are positive about your paper (although as you will see, there is some suggestion that it might be appropriate to make clear that your data are somewhat preliminary) and I am very happy to provisionally accept it for publication subject to revision. As you will see, our reviewers make a number of comments that I would like you to address as you prepare a final version. Thanks for choosing PeerJ as an outlet for your work, and congratulations on a very interesting paper!

Reviewer 1 ·

Basic reporting

No Comments

Experimental design

This study by Han, Martin et al. investigates the potential influence of maternal premastication of shared food on the developing infant salivary microbiome. The data indicate little-to-no influence of this behavior on the oral bacterial community of the infant, however this study is well-conducted and mostly methodologically sound. The authors are also to be commended for their refreshing straightforwardness regarding the limitations of the study and their appropriately narrow conclusions from the microbiome data. The topic is relevant and meaningful, and it is the opinion of this reviewer that the manuscript merits publication (in principle), especially given the difficulty of obtaining these very valuable types of samples and the intriguing hypotheses the authors test.

Validity of the findings

I recommend a minor revision to address a few methodological and analytical concerns.

Major Issues

It is atypical in the field to trust ~250 bp 16S amplicons clustered at 97% OTU similarity to provide enough taxonomic information to accurately describe members of the microbial community at a species level, as the authors have done here. (See Liu et al, Nucleic Acids Research 2008 and https://peerj.com/preprints/934/ for details). I recommend presenting only genus and higher-level data, especially given the lack of sequenced cultured bacterial isolates from the samples, metagenomic sequencing, or some other confirmatory method for the taxonomic assignments. This will impact the author’s conclusions regarding the presence of specific potential pathogens.

I am unsure of the absolute level of influence of their metadata (infant vs. mother, dyad number, etc.) on the bacterial community structure, as it may be small or not statistically significant. To quantify the degree to which the metadata on the samples influences bacterial community structure I recommend an ANOSIM, PERMANOVA (adonis), or some similar approach be used on the distance matrices and the strength and statistical significance of the influence be reported.

Minor Issues

While the main conclusions made here are likely robust to sequencing depth, the authors may wish to avoid rarefying their data so drastically in the future (other than for alpha diversity measures). Going from 532,825 sequences in a sample to 3,148 sequences is a choice that loses a lot of information, and alternative normalization methods may improve the sensitivity of their tests (see McMurdie and Holmes PLoS Computational Biology 2014). Similarly, I am unclear if the Student’s t-test statistical analysis for differential taxa abundance significance was performed on the rarefied raw counts or on taxa proportions. Were the data normally distributed as t-tests assume? To avoid these problems, I would suggest instead that one of the dedicated software packages for this be used (such as PhyloSeq, ANCOM, MetagenomeSeq, or ALDEx2), which may also better handle the overdispersion, compositionality, sparsity, and heteroscedasticity typical to sequencing data.

While the authors correctly list the differences in DNA extraction methodology as a limiting factor in their study, understanding the exact extraction method used is critical for future cross-comparison of studies. I am unclear on the details of the methods used here. It appears the PSP Saliva Gene Kit is not ideal for lysis of many Gram positive species, as there is apparently no bead beating step. If true, the authors should mention this fact. For the food samples, which Powerlyzer kit (there are multiple) did the authors use?

As a potential confounding factor, do individuals other than the mother premasticate food for the infants? A statement containing any known information on this question should be added..

If the data is available, it would be nice to investigate if the amount of breastfeeding an infant received plays a part in determining the infant oral microbiome.

Additional comments

In line 381: “In subsampling 3148 sequences, P. gingivalis was observed in 5 mothers’ saliva samples (41.7%, MSA1, MSA8, MSA12, MSA13, MSA15), but not in any of the infants’ samples (41.7%, p=0.0009)” why is the second “41.7%” there?

·

Basic reporting

The introduction contains appropriate information and is generally written clearly. I am not as familiar with the oral microbiome literature, but relevant sources appear to be cited. I suggest some minor changes to flow and structure below in "general comments." Figures are appropriate and the indication is that raw data will be shared.

Experimental design

The experimental design is generally sufficient. My major concerns are the small sample size and variation in sample collection and processing. The authors address these concerns in the discussion, but in a few cases more specificity would be helpful (i.e. how big an effect are these variables likely to have?). Do we have any idea how big an effect we would expect in this situation and whether the sample size is sufficient or the confounds are likely to interfere? I was also unclear on when premastication actually occurred in the sampled individuals (how recent is "recent" and how variable was this?), and I did not understand the food sampling protocol and would like clarification in that section (as described below). Finally, the authors' description of choosing only unique sequences to analyze seemed odd to me since it appeared that they were limiting their data to presence-absence data. Since the data are not presented this way, I think I am misunderstanding, but the text should be clarified.

Validity of the findings

A reorganization of the results would be helpful to improve clarity. That being said, the authors make appropriate conclusions based on this small dataset with potentially confounding variables. However, it might be useful to use the word "preliminary" or something similar in the discussion somewhere. The last section of the discussion is interesting speculation, but I think the ideas need to tied together more smoothly to strengthen the message. I would also like to see the authors briefly revisit the topic of adult dental hygiene and potential impacts on their comparisons with other populations as well as the maternal-infant transmission in this population. Also, do we think the type of food being shared will impact these transmission pathways? Or would we expect it all to be the same? Speculation, I realize, but perhaps interesting. Lastly, some additional discussion regarding frequency of premastication would be appropriate. The authors discuss infant developmental stage and breasfeeding, but frequency of premastication may affect these transmission relationships and didn't seem to be well-accounted for here. Therefore, in such a small, uncontrolled sample, it may be difficult to detect a signal.

Additional comments

These data are novel, and the findings are intriguing. As the authors state, additional research with larger sample sizes and more consistent methodology are necessary. However, this is a nice preliminary foray into the topic. My detailed comments are as follows:

Line 68: “has been” instead of “was”
Line 126: I suggest a transitional sentence or word here. Just something like “Therefore, to XYZ, we collected” would work. Actually, I don’t think the details about sample collection are necessary here since it is covered in the methods.
Line 144: What does “recent” mean?
Line 171: Were subjects in Villages 1 and 2 sampled before food consumption as well?
Line 188: I’m not clear on what the controls are…how are they controls if they are also chewed? Also, here it says there are five controls, but in line 204 it says there are two. I am also not clear how FD14 and FD12 fit in. This appears to match the number of controls, but they are listed as pre-masticated foods in Table 1…
Line 207: I think this may explain my confusion above. I suggest making the previous description of samples and controls clearer and/or more comprehensive and perhaps listing the samples that dropped out in this section.
Line 231-2: Why were unique sequences chosen? Results are presented on relative abundance, but relative abundances are likely to be biased if only unique sequences were considered…(i.e. if three of the same sequence were present in the data, it would only be counted once?)
Line 251-3: Were relative abundances distributed normally? Often this is not the case, especially at the OTU level so using a parametric test is not usually advisable.
Line 275-287: Move to methods
Line 319-336: This information seems to be more general and could be included in the previous results subsection. The same is true of the last paragraph in this subsection. Or perhaps the heading should be changed? Additionally, it might be useful to describe general characteristics of mother-infant microbial communities, then the relationship between premasticated food and maternal microbiota, and then discuss whether relatedness/premastication influences infant microbial communities… And the final section about pathogens would flow nicely from this.
Line 431: Briefly state how big a confound this is likely to be.
Line 464: Be a little more specific.
Line 470: italics, were these Streptococcus observed in this study?
Line 477: I wonder about maternal oral health in this population compared to the others referenced in throughout the paper in a comparative sense. It would be helpful to have some brief information somewhere about dental hygiene in the Tsimane and how it differs from other previously-studied populations and how that may or may not affect maternal-infant transfer. The authors allude to this in the introduction, but it might be worth explicitly mentioning in the discussion.
Lines 467-488: I suggest tying these ideas together more clearly/directly.
Figure 1B: No index numbers with food samples?

Reviewer 3 ·

Basic reporting

The paper is well-written, well anchored to existing literature, and provides exciting information about the dynamics of oral microbiome of mother and infants in the context of a common cultural behavior- premastication. Greatly enjoyed reading this article and look forward to citing it and featuring it in my classes. No comments toward improvement.

Experimental design

The experimental design is precisely described, ethical designed, and foundational for replication in the same and other populations. No comments toward improvement.

Validity of the findings

Results represent a robust evaluation of similarity or differences between mothers and infants, within and across dyads. The statistical framing is largely strong. One recommendation: Line 354-355 “more similar to those of their donors than to those of other mothers, although this finding only trends toward statistical significance” provide the p-value, but just say “not significant.” https://mchankins.wordpress.com/2013/04/21/still-not-significant-2/

Additional comments

Very interesting paper, thank you for studying these under-researched topic.

Minor comments:

Line 113, replace “birthed” with “delivered”

Line 25, were these data presented in Dr. Martin’s dissertation anywhere? Could the dissertation be cited here?

Line 149-150 “Each subject’s age (in years), number of teeth, and number of caries (observed by Author 2) are 150 reported in Table 1.” I recommend not having a topic sentence that effectively points the reader away from the paragraph. Greater communicative value, and crisper writing will be “Mothers and infants varied in their respective ages and dental health (Table 1)”

Line 275-276 is again a topic sentence that directs the reader away from the content of the paragraph. Recommend moving sentence to later or end of the paragraph.

Line 293-294: recommend minor edit “The relative abundance of each phylum was different between the saliva and food samples of mothers and infants (Figures 1A & 1B).”

Line 483: discussion of infant similarity in oral microbiome: cite in vivo study of antimicrobial interaction between breast milk & saliva: Al-Shehri, Saad S., et al. "Breastmilk-Saliva Interactions Boost Innate Immunity by Regulating the Oral Microbiome in Early Infancy." PloS one 10.9 (2015): e0135047.

---

## Round 0.2 · accepted · Accept

Dear Joe, Cliff and Melanie,

Thanks very much indeed for your revision, and the detailed rebuttal. I'm now very happy to accept your paper for publication, and thank you very much indeed for considering PeerJ as an outlet for your work. The findings are novel and interesting, and I'm sure they'll generate a lot of interest.

Congratulations, and all the best,

Lou